# Gap junctions composed of connexins 41.8 and 39.4 are essential for colour pattern formation in zebrafish

Uwe Irion*, Hans Georg Frohnhöfer, Jana Krauss[†], Tuğba Çolak Champollion[‡], Hans-Martin Maischein[§], Silke Geiger-Rudolph, Christian Weiler, Christiane Nüsslein-Volhard

Max Planck Institute for Developmental Biology, Tübingen, Germany

**Abstract** Interactions between all three pigment cell types are required to form the stripe pattern of adult zebrafish (*Danio rerio*), but their molecular nature is poorly understood. Mutations in *leopard* (*leo*), encoding Connexin 41.8 (Cx41.8), a gap junction subunit, cause a phenotypic series of spotted patterns. A new dominant allele, *leo^{tK3}*, leads to a complete loss of the pattern, suggesting a dominant negative impact on another component of gap junctions. In a genetic screen, we identified this component as Cx39.4 (*luchs*). Loss-of-function alleles demonstrate that *luchs* is required for stripe formation in zebrafish; however, the fins are almost not affected. Double mutants and chimeras, which show that *leo* and *luchs* are only required in xanthophores and melanophores, but not in iridophores, suggest that both connexins form heteromeric gap junctions. The phenotypes indicate that these promote homotypic interactions between melanophores and xanthophores, respectively, and those cells instruct the patterning of the iridophores.

*For correspondence: uwe. irion@tuebingen.mpg.de

Present address: [†]Auengrund 7, Schönau-Berzdorf, Germany; [‡]Skirball Institute of Biomolecular Medicine, New York University, Langone Medical Center, New York, United States; [§]Max Planck Institute for Heart and Lung Research, Bad Nauheim, Germany

**Competing interests:** The authors declare that no competing interests exist.

**Reviewing editor**: Marianne E Bronner, California Institute of Technology, United States

## Introduction

Adult zebrafish (*Danio rerio*) display a characteristic pattern of horizontal dark and light stripes on their bodies as well as on their anal fins and tailfins (*Figure 1A*). Three types of pigment cells (chromatophores) are required to create this pattern. In the dark stripes of the trunk, a net of loose or blue iridophores and pale, stellate xanthophores cover the melanophores, the light stripes are composed of dense silvery iridophores covered by compact orange xanthophores (*Hirata et al., 2003*, *2005*; *Frohnhofer et al., 2013*; *Mahalwar et al., 2014*; *Singh et al., 2014*).

The adult pigmentation pattern is formed during metamorphosis, a period between approximately 3 and 6 weeks of development. At the onset of metamorphosis, iridophores appear in the skin at the region of the horizontal myoseptum that provides a morphological pre-pattern (*Frohnhofer et al., 2013*). They proliferate and spread as densely connected cells to form the first light stripe. While spreading further ventrally and dorsally, into the regions where the first two dark stripes will form, they change their appearance and become more loose, then they aggregate again at a distance to form the next light stripes (Singh et al., 2013). Larval xanthophores, covering the flank of the fish, start to proliferate and re-organize into densely packed compact cells above the dense iridophores of the light stripe and into more loosely organized, stellate cells in the dark stripe regions (*Mahalwar et al., 2014*). Melanoblasts migrate along spinal nerves into the skin in the presumptive stripe regions where they finally differentiate and expand to fill the space (*Budi et al., 2011*; *Dooley et al., 2013*; *McMenamin et al., 2014*; *Singh et al., 2014*). In the fins, stripe formation does not require iridophores, suggesting that the patterning mechanisms in the body and fins are different.

A number of mutants are known in which the pattern is not formed normally. In one class of them, one type of pigment cell is absent; in mutants for *nacre/mitfA* melanophores are missing (*Lister et al., 1999*),

**eLife digest** The colour patterns that mark an animal's skin, hair, or feathers—called the pigmentation pattern—can be very important for its survival and fitness, helping it to hide from predators or to attract a mate. As a result, there is considerable interest in understanding how genes, proteins, and cells work together to produce the many different pigmentation patterns that exist in the animal world.

Adult zebrafish have a characteristic pigmentation pattern of horizontal dark and light stripes on their bodies and fins. There are three types of pigment cell that create this pattern. Xanthophores and iridophores are found all over the body, and the dark stripes also contain melanophore cells. The silvery, reflective iridophores are the first of the cells to populate the skin, giving rise to the first light stripe. They then form a dense network of cells that breaks up to form the darker stripes. However, iridophores are not required to form stripes in the fins, suggesting that patterning occurs differently in the fins and the body.

Mutations to a gene called *leopard*, or *leo* for short, cause spots to form on the skin of the zebrafish in place of the usual stripes. This gene encodes a member of the connexin family of proteins, which form channels in the membranes that surround cells. These channels—known as gap junctions—allow neighbouring cells to communicate with each other. Each gap junction is made up of two half channels, with one half coming from each neighbouring cells. If the two half channels are identical, the gap junction is known as 'homomeric'; 'heteromeric' gap junctions are made from two different half channels, each consisting of a different connexin protein. The connexin encoded by *leo* is required for both types of gap junction to form between melanophores and xanthophores.

Irion et al. discovered a new mutation to the *leo* gene that completely disrupts the patterning of the zebrafish. A technique called a genetic screen revealed that the same patterning defects are also seen in the body of zebrafish with mutations to another gene called *luchs*, which encodes a different connexin protein to the one produced by *leo*. However, the fins of zebrafish with mutant versions of *luchs* remain striped.

The findings of Irion et al. suggest that heteromeric gap junctions formed from the connexins produced by *leo* and *luchs* are important for xanthophores and melanophores to communicate with each other and so form the stripy patterning seen on the body of the zebrafish. The signals transmitted through the gap junctions may also make the iridophores adopt the looser arrangement that is required for the dark stripes to form. As a next step, it will be important to identify the signals that pass through these gap junctions that allow the cells to communicate with their neighbours and establish the pigmentation pattern.

mutations in *pfeffer/csf1rA*, lead to the lack of xanthophores (*Odenthal et al., 1996*; *Parichy et al., 2000b*), and in *shady/ltk*, *rose/ednrb1b*, and *transparent/mpv17* mutants iridophores are absent or strongly reduced (*Parichy et al., 2000a*; *Lopes et al., 2008*; *Krauss et al., 2013*). In all these cases, the remaining two types of chromatophores form an irregular residual striped pattern. These genes are autonomously required in the respective cell types indicating that interactions among all three chromatophore types are necessary to generate the striped pattern on the trunk of the fish (*Maderspacher and Nusslein-Volhard, 2003*; *Parichy and Turner, 2003*; *Frohnhofer et al., 2013*; *Krauss et al., 2014*). Based on the analysis of these mutants and on ablation experiments, several attractive and repulsive signals acting over long or short ranges between the chromatophores have been postulated (*Maderspacher and Nusslein-Volhard, 2003*; *Yamaguchi et al., 2007*; *Nakamasu et al., 2009*; *Frohnhofer et al., 2013*; *Patterson and Parichy, 2013*; *Krauss et al., 2014*).

In another class of mutants, an abnormal pattern is formed with all three chromatophore types present (*Haffter et al., 1996*); in these animals, the communication between the cells might be affected. The genes identified in this group encode integral membrane proteins, for example, *obelix/Kir7.1*, a rectifying potassium channel (*Iwashita et al., 2006*), *seurat/Igsf11*, a cell-adhesion molecule of the immunoglobulin superfamily (*Eom et al., 2012*), or *dali/Tetraspanin 3c*, a transmembrane-scaffolding protein (Inoue et al., 2012). The best-known example for this class of mutants is *leopard* (*leo*), where the stripes are wavy or broken up into a series of dark spots (*Figure 1B*). The original mutant has been regarded as a separate *Danio* species (*Kirschbaum, 1975*; *Kirschbaum, 1977*; *Frankel, 1979*).

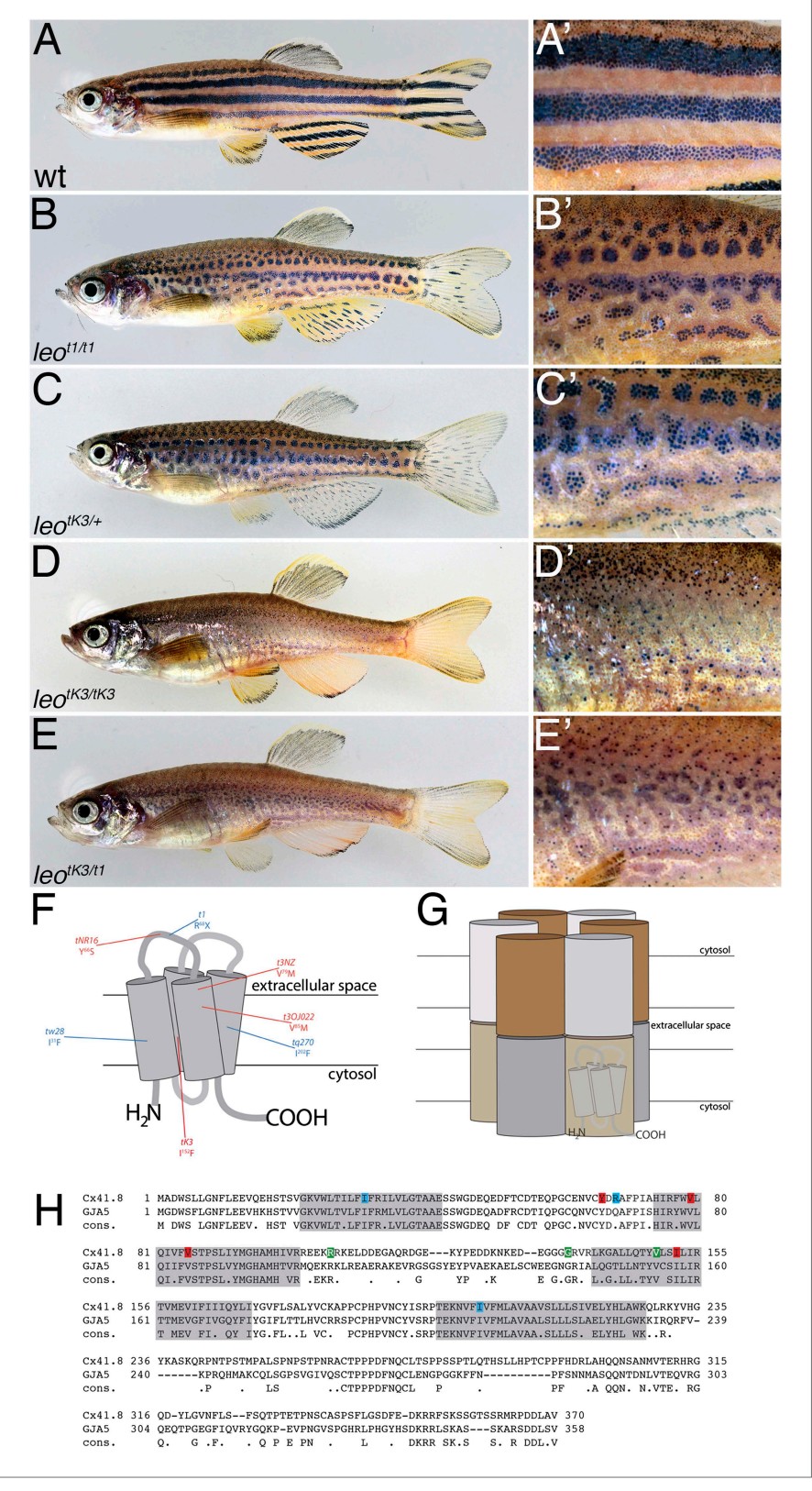

**Figure 1**. The *leo* mutant phenotype. Wild-type zebrafish (**A**) show a pattern of dark and light stripes on the body and on anal- and tail-fins. At higher magnification (**A'**), dark melanophores in the stripe regions and orange xanthophores in the light stripe regions are discernible. In mutants homozygous for *leo^t1* (**B**) and heterozygous for
*Figure 1. Continued on next page*

*Figure 1. Continued*

*leo^{tK3}* (**C**), the stripes are dissolved into spots. Clusters of melanophores are still visible (**B'** and **C'**). Fish homozygous for *leo^{tK3}* (**D**) or trans-heterozygous for *leo^{tK3}* over *leo^{t1}* (**E**) show an identical phenotype of a completely dissolved pattern. Individual melanophores that hardly cluster together are still present, mostly associated with blue iridophores (**D'** and **E'**). In (**F**) a cartoon of Connexin 41.8 is depicted showing the positions of the *leo* mutations. Gap junctions are composed of two hemi-channels in adjacent cells. Each hemi-channel is made of six connexin subunits, they can be identical (homomeric) or different (heteromeric). In (**G**) a heteromeric/heterotypic gap junction is schematically shown. An alignment of the amino acid sequence from zebrafish Cx41.8 with its human orthologue, GJA5, is shown in (**H**). The transmembrane regions are shaded in grey. Newly identified point mutations in Connexin 41.8 are highlighted in red: *leo^{tNR16}*: Y$^{66}$S; *leo^{tNZ}*: V$^{79}$M; *leo^{t3OJ022}*: V$^{85}$M; *leo^{tK3}*: I$^{152}$F. Already known alleles are highlighted in blue: *leo^{tw28}*: I$^{31}$F; *leo^{t1}*: R$^{68}$X; *leo^{tq270}*: I$^{202}$F. Polymorphisms found in sequences from wild-type fish are highlighted in green: 106 R/K, 136 G/R; 149 V/I.

Subsequently several dominant alleles were identified in *Danio rerio* (***Haffter et al., 1996***), and it has been shown that the *leo* phenotype is caused by a mutation in *connexin 41.8* (***Watanabe et al., 2006***), which codes for a subunit of gap junctions (gap junction protein α5, GJA5).

Gap junctions are intercellular channels that allow the passage of small molecules and ions between neighbouring cells, and thus are responsible for their chemical and electrical coupling (***Kar et al., 2012***). They are formed by the juxtaposition of two hemi-channels (connexons), composed of six connexin subunits, in adjacent cells (***Unwin and Zampighi, 1980***). Connexins are integral membrane proteins with four transmembrane domains, two extracellular loops, one intracellular loop, and intracellular N- and C-termini (***Milks et al., 1988***) (***Figure 1F,G***). Hemi-channels can be composed of different subunits; the resulting gap junctions are homotypic, if identical connexons from neighbouring cells pair, or heterotypic, if different hemi-channels come together. In addition to the subunit composition, gap junction conductivity is regulated by a number of different factors, for example, by the intracellular levels of Ca$^{2+}$, by polyamines, by the membrane potential, or by phosphorylation (***Thévenin et al., 2013***).

It has been reported that the function of the *leo* gene is required in two of the chromatophore types, the melanophores and the xanthophores, for homotypic and heterotypic cellular interactions (***Maderspacher and Nusslein-Volhard, 2003***). Wild-type and mutant forms of Cx41.8 artificially expressed in melanophores can lead to different variations of the stripe pattern (***Watanabe and Kondo, 2012***). This could indicate that gap junctions are responsible for some of the short range signals postulated to occur within and between these two chromatophore types (***Parichy and Turner, 2003***; ***Inaba et al., 2012***; ***Frohnhofer et al., 2013***).

Three alleles of *leo* have been described so far. The original allele, *leo^{t1}*, is recessive; it has a premature stop codon at position 68 of the coding sequence and is most likely a functional null-allele. Two dominant alleles, *leo^{tq270}* and *leo^{tw28}*, carry missense mutations; they lead to a stronger phenotype (***Haffter et al., 1996***; ***Watanabe et al., 2006***). The dominance of these alleles led to the suggestion that heterotypic as well as homotypic connexons containing Cx41.8 could be involved in pigment patterning, postulating the existence of other connexin partner(s) in the potential heterotypic channels (***Watanabe et al., 2006***).

In this study, we identified several additional alleles of *leo*. All of them are dominant, and the strongest of them, *leo^{tK3}*, leads to a complete loss of the pigmentation pattern in homozygous carriers; the number of melanophores is reduced, and they appear as small groups or individual cells in the skin, embedded in an expanded light region of dense iridophores covered by xanthophores. In a genetic screen, we found two dominant alleles of a gene we named *luchs* (*luc*) as enhancers of the *leo* loss-of-function phenotype. We show that the *luc* gene codes for another connexin, Cx39.4. A loss-of-function allele for *luc* displays a phenotype similar to *leo^{t1}*; however, the fins are not affected. Further we show that both genes, *leo* and *luc*, are required in the trunk in xanthophores and in melanophores, but not in iridophores. Our results suggest that Cx41.8 and Cx39.4 form heteromeric gap junction channels in the plasma membranes of melanophores and xanthophores. In the complete absence of the channels, iridophores take over and almost fully fill the space normally occupied by alternating light and dark stripes, whereas melanophores are suppressed. This suggests that the heteromeric gap junctions function in the communication between xanthophores and melanophores and potentially in the transduction of signals to the dense iridophores to induce the transition into the loose shape required for dark stripe formation.

## Results

### New alleles of *leo*

In a number of experiments designed to find mutants with adult phenotypes, we identified four additional alleles of *leo*. All of them are dominant. Three of our new alleles result in a phenotypic series of patterns from undulating stripes to breaking up of the dark stripe regions into small spots. As in wild-type (*Figure 1A–A'*), the dark stripe regions of melanophores are covered with loose blue iridophores. The light stripe regions composed of dense iridophores covered by compact xanthophores are expanded and ingress into the stripe regions. Depending on the genotypes, they display varying strengths, similar to the previously described dominant alleles *leo^tw28* and *leo^tq270* (*Haffter et al., 1996*; *Watanabe et al., 2006*). However, one allele, *leo^tK3*, is considerably stronger. In heterozygous fish, it leads to an intermediate phenotype similar to the one seen in homozygotes for the loss-of-function allele *leo^t1*. Fish homozygous for *leo^tK3*, or trans-heterozygous for *leo^tK3* and *leo^t1*, show a much stronger phenotype with a loss of any striped arrangement (*Figure 1B–E*). Instead of dark and light stripe regions, they display variable numbers of small groups of melanophores in islands of blue iridophores, or even single melanophores distributed in an almost even background of dense iridophores covered by xanthophores characteristic for the light stripe regions (*Figure 1D',E'*). The stripes in the anal and caudal fins are also affected in *leo* mutants, weak phenotypes show some short residual melanophore stripes, in homozygous *leo^tK3* animals the pattern is completely lost and only few melanophores are present at the margins of the fins.

The newly identified *leo* alleles carry mis-sense mutations in the coding sequence for Connexin 41.8. The affected amino acid residues are at highly conserved positions in the N-terminal half of the protein (*Figure 1F,H*); the mutation in *leo^tK3*, Ile$^{152}$Phe, lies within the third transmembrane domain at a position where all connexins invariably have a small hydrophobic residue.

### Dominant mutations in *luc* enhance the phenotype of *leo^t1*

The fact that the mis-sense mutation in *leo^tK3* leads to a much stronger phenotype than the one seen in fish homozygous for the loss-of-function allele *leo^t1* argues for a dominant negative effect of the mutant protein on an additional component involved in pigment patterning. To identify this component, we performed a genetic screen for dominant mutations that enhance the phenotype of *leo^t1*. We mutagenized males homozygous for *leo^t1* with ENU and crossed them to homozygous mutant females. In their offspring, we screened for fish with a stronger phenotype, similar to the one observed in *leo^tK3* mutants. Among 4469 F1 fish, we found three individuals with a strongly enhanced phenotype. One of them turned out to be an allele of *obelix* (*Haffter et al., 1996*), which has been shown before to enhance the *leo* phenotype (*Maderspacher and Nusslein-Volhard, 2003*). The other two mutants, *tXA9* and *tXG1* (*Figure 2B,E*), carry alleles of the same gene, which we named *luchs* (*luc*) after the German name for lynx. Both have almost identical phenotypes, heterozygous fish show wavy stripes with some gaps, in homozygotes and in trans-heterozygotes, the stripes are completely dissolved into very small spots and individual melanophores (*Figure 2C,D,F,G*). Mutants of *luc* display patterns that are not significantly different from those of the *leo* phenotypic series, however, the fins are much less affected in our new mutants. To identify the responsible mutations, we used a candidate approach and sequenced the coding sequences of several *connexin* genes. For both alleles we identified a mis-sense mutation in the gene encoding Connexin 39.4 (*cx39.4*). In *luc^tXA9* an A to C transversion leads to an amino acid exchange T$^{29}$P, in *luc^tXG1* a G to T transversion results in W$^{47}$L, both residues are highly conserved and lie within the first transmembrane domain and the first extracellular loop, respectively (*Figure 3A*).

### *luc* function in pigment patterning

To investigate the role of Cx39.4 in pigment pattern formation in zebrafish, we created loss-of-function mutations in the *luc* gene using the CRISPR/Cas9 system (*Hwang et al., 2013*). We targeted the 5' region of the coding sequence (*Figure 3B*) and found a high incidence of small deletions leading to frame shift mutations. Most often we found a 3 bp (CTC) deletion accompanied by a 1 bp (G) insertion resulting in a premature stop codon and leading to a truncation of the translated protein after 18 amino acids. We also frequently found a 22 bp deletion resulting in a truncated protein of only 13 amino acids followed by seven unrelated residues (*Figure 3C*). Fish homozygous or trans-heterozygous for these knock-out alleles (*luc^k.o.*) develop an irregular pattern with interrupted stripes and spots very

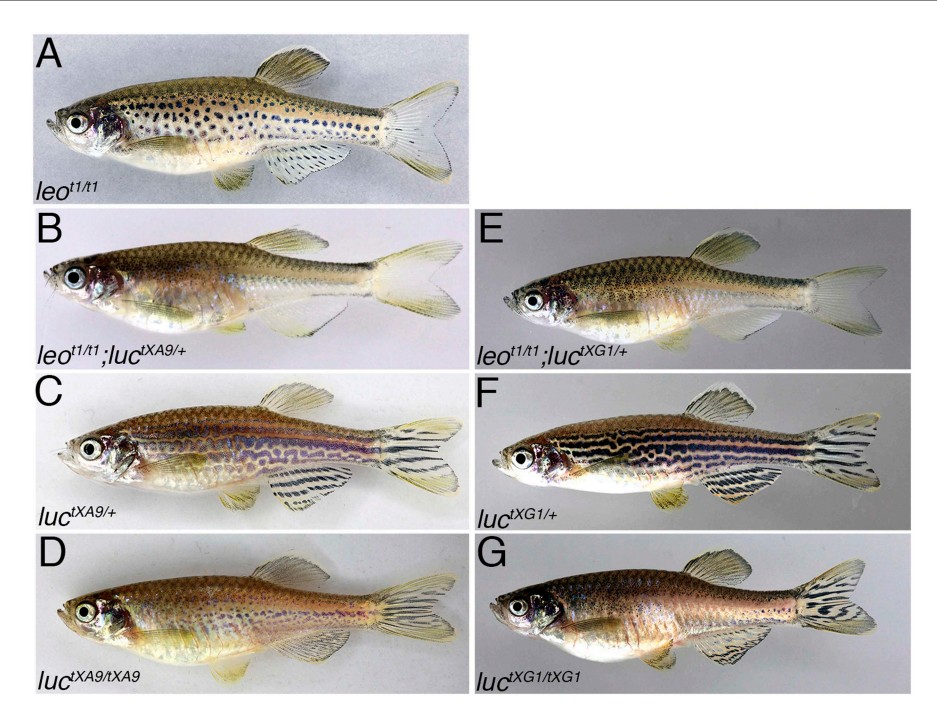

**Figure 2**. Dominant enhancers of the *leo^t1* phenotype. The spotted pattern of fish homozygous for *leo^t1* (**A**) is lost when they carry an additional mutant allele for *luc^tXA9* (**B**) or *luc^tXG1* (**E**). Fish heterozygous for *luc^tXA9* (**C**) and *luc^tXG1* (**F**) have undulating and disrupted stripes on their flanks. In mutants homozygous for *luc^tXA9* (**D**) or *luc^tXG1* (**G**) the pattern is almost as strongly disrupted as in *leo^tK3* mutants (compare to **Figure 1D**).

similar to *leo* mutants. Again, like in the case of the dominant alleles, the patterning of the fins is less affected in the mutants and they show an almost normal arrangement of stripes (**Figure 3D**). We also reverted the phenotype of heterozygous *luc^tXA9* mutants to wild-type by inducing loss-of-function mutations with the CRISPR/Cas system on the chromosome carrying the dominant allele. This confirms that the *luc* alleles carry mutations in the Cx39.4 gene. Interestingly, double mutants homozygous for loss-of-function alleles of *leo* and *luc* display a phenotype which is indistinguishable from that of homozygous mutants of the dominant *leo^tK3* allele and slightly stronger than that of the homozygous dominant alleles of *luc* (**Figure 3E**). This indicates that the dominant negative effect observed in *leo^tK3* is mediated through connexin 39.4, and that in *leo^tK3* both the *leo* and *luc* functions are abolished. Our results are most easily explained if the two connexins form heteromeric gap junctions.

## The development of the *leo* and *luc* phenotypes

To establish the time point when the phenotype of the *leo* and *luc* mutants becomes first apparent, we followed wild-type and mutant fish during the course of metamorphosis. The first signs of the phenotype become visible at stage PR (8.6 mm SSL). When in wild-type, melanophores gather immediately dorsally and ventrally to the dense iridophores of the first light stripe, they stay more dispersed in the mutants (**Figure 4A–E**). Dense iridophores covered with xanthophores disperse in an irregular manner in the mutants to form a wider first light stripe, branching into the dark stripe region and suppressing melano-phores dorsally and ventrally to form irregular patches (**Figure 4F–J**). At stage J++, when in wild-type three dark and three light stripes are formed, in the *luc* and *leo* loss-of-function mutants the spotted pattern is already clearly apparent (**Figure 4L,M**) and the dominant mutants show fields of dense iridophores and xanthophores with few, scattered melanophores (**Figure 4N,O**). This analysis reveals that the *leo;luc* function is required during development and the mutant patterns are not caused by pattern instability.

## Requirement of *leo* and *luc* function in chromatophores

To understand the function of *leo* and *luc* in the different types of chromatophores, we analysed com-binations with mutants lacking one of the three chromatophore types, in which incomplete residual

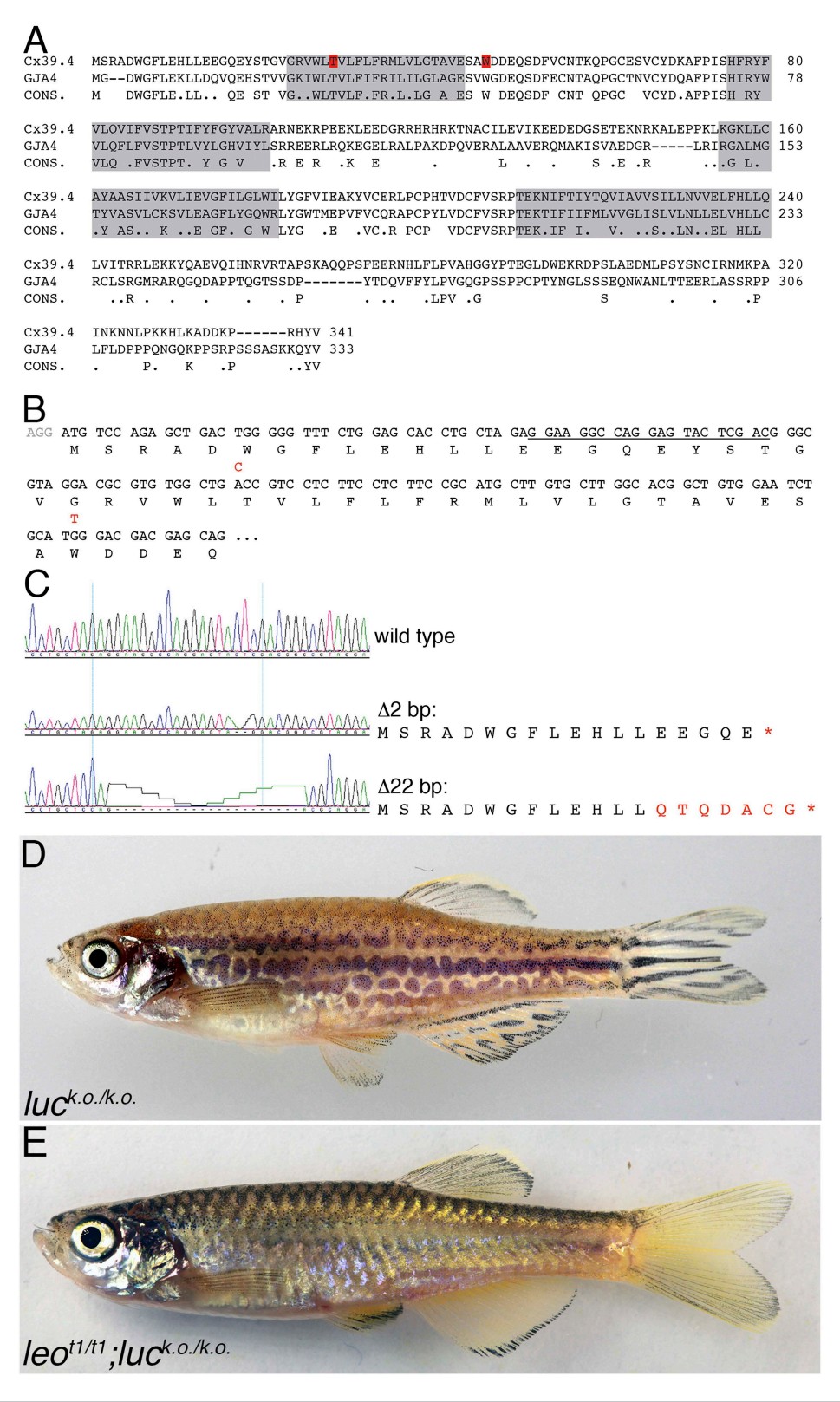

**A**

```
Cx39.4  MSRADWGFLEHLLEEGQEYSTGVGRVWLTVLFLFRMLVLGTAVESAWDDEQSDFVCNTKQPGCESVCYDKAFPISHFRYF  80
GJA4    MG--DWGFLEKLLDQVQEHSTVVGKIWLTVLFIFRILILGLAGESVWGDEQSDFECNTAQPGCTNVCYDQAFPISHIRYW  78
CONS.   M   DWGFLE.LL.. QE ST VG..WLTVLF.FR.L.LG A ES W DEQSDF CNT QPGC  VCYD.AFPISH RY

Cx39.4  VLQVIFVSTPTIFYFGYVALRARNEKRPEEKLEEDGRRHRHRKTNACILEVIKEEDEDGSETEKNRKALEPPKLKGKLLC  160
GJA4    VLQFLFVSTPTLVYLGHVIYLSRREERLRQKEGELRALPAKDPQVERALAAVERQMAKISVAEDGR-----LRIRGALMG  153
CONS.   VLQ .FVSTPT. YG V  .RER .KE  .     .     L. . S.E .R    ...G L.

Cx39.4  AYAASIIVKVLIEVGFILGLWILYGFVIEAKYVCERLPCPHTVDCFVSRPTEKNIFTIYTQVIAVVSILLNVVELFHLLQ  240
GJA4    TYVASVLCKSVLEAGFLYGQWRLYGWTMEPVFVCQRAPCPYLVDCFVSRPTEKTIFIIFMLVVGLISLVLNLLELVHLLC  233
CONS.   .Y AS.. K ..E GF. G W LYG  .E  .VC.R PCP  VDCFVSRPTEK.IF I.  V. ..S.LN..EL HLL

Cx39.4  LVITRRLEKKYQAEVQIHNRVRTAPSKAQQPSFEERNHLFLPVAHGGYPTEGLDWEKRDPSLAEDMLPSYSNCIRNMKPA  320
GJA4    RCLSRGMRARQGQDAPPTQGTSSDP-------YTDQVFFYLPVGQGPSSPPCPTYNGLSSSEQNWANLTTEERLASSRPP  306
CONS.   ..R. . .  .P    . ..  .LPV .G        S   .  .P

Cx39.4  INKNNLPKKHLKADDKP------RHYV  341
GJA4    LFLDPPPQNGQKPPSRPSSSASKKQYV  333
CONS.   .    P. K  .P      ..YV
```

**B**

```
 AGG ATG TCC AGA GCT GAC TGG GGG TTT CTG GAG CAC CTG CTA GAG GAA GGC CAG GAG TAC TCG ACG GGC
     M   S   R   A   D   W   G   F   L   E   H   L   L   E   E   G   Q   E   Y   S   T   G
                                                 C
GTA GGA CGC GTG TGG CTG ACC GTC CTC TTC CTC TTC CGC ATG CTT GTG CTT GGC ACG GCT GTG GAA TCT
 V   G   R   V   W   L   T   V   L   F   L   F   R   M   L   V   L   G   T   A   V   E   S
     T
GCA TGG GAC GAC GAG CAG ...
 A   W   D   D   E   Q
```

**C**

wild type

Δ2 bp:
M S R A D W G F L E H L L E E G Q E *

Δ22 bp:
M S R A D W G F L E H L L Q T Q D A C G *

**D**

*luc*^k.o./k.o.

**E**

*leo*^t1/t1;*luc*^k.o./k.o.

**Figure 3**. *luc* loss-of-function leads to patterning defects. A comparison of the amino acid sequences of Cx39.4 of zebrafish, encoded by the *luc* gene, and human Cx37 (GJA4) is shown in (**A**). The transmembrane domains are shaded in grey, the residues mutated in *luc*^tXA9 (T^29 P) and *luc*^tXG1 (W^47 L) are highlighted in red. Note the extension of

*Figure 3. Continued on next page*

*Figure 3. Continued*

the N-terminal cytoplasmic domain of Cx39.4 by two amino acid residues. In (**B**) the beginning of the coding sequence for Cx39.4 is shown, the mutations in *luc^tXA9* and *luc^tXG1* are indicated in red above the sequence; the CRISPR target site is underlined. The sequence traces of wild-type fish and two fish homozygous for deletions of 2 and 22 bp are shown in (**C**). The predicted amino acid sequences for these mutants are indicated. In (**D**) a fish homozygous for the 2 bp deletion is shown, the striped pattern on the flank of the fish is disrupted and partly dissolved into spots, the fins are almost normal. Double mutants for *leo^t1* and *luc* loss-of-function (**E**) have only very few melanophores on the flank and show an almost uniform pattern of iridophores and xanthophores.

striped patterns are generated (*Maderspacher and Nusslein-Volhard, 2003*; *Frohnhofer et al., 2013*). We created double mutants with *leo^tK3*, where *leo* and *luc* functions are abolished, and observed that the residual striped pattern is lost in all three cases (*Figure 5*). In *nac/mitfa* mutants, where melanophores are absent, light stripe fields of dense iridophores covered by xanthophores are enlarged and display irregular boundaries between regions of blue iridophores characteristic for the dark stripe (*Figure 5A,A'*). Double mutants of *nac* with *leo^tK3* show an expansion of dense iridophores covered with xanthophores, in some cases blue iridophore regions are completely lost (*Figure 5B*). In *pfe/csf1rA* homozygotes, which lack xanthophores, melanophore stripes are broken into rows of spots separated by dense iridophores, and isolated melanophores appear in the light stripe regions (*Figure 5C,C'*). If *pfe* is combined with *leo^tK3*, individual isolated melanophores are evenly dispersed over the flank of the fish that are almost uniformly covered with dense iridophores (*Figure 5D*). In mutants lacking iridophores, *tra/mpv17*, *shd/ltk,* or *rse/ednrb1b*, the first two melanophore stripes are broken into spots, and further stripes are missing (see *Figure 5E,E'* for *shd* and *Figure 5—figure supplement 1G,G'* for *tra*). In the double mutants with *leo^tK3*, only few, if any, melanophores are present that are scattered without forming an aligned pattern (*Figure 5F*). We also analysed these double mutant combinations with *leo^t1* or *luc^k.o.* (*Figure 5—figure supplement 1*) and observed very similar, albeit slightly weaker double mutant phenotypes.

In the mutants, the strongest phenotypic effect is seen in iridophores, which seem to appear mostly in the dense form, representative of the light stripes. However, iridophores do not autonomously require the *leo* or *luc* function (see below, *Figure 6A,A'*). This suggests that iridophore behaviour is affected via xanthophores and/or melanophores. From the *pfe* phenotype an inhibitory effect of xanthophores on the expansion of dense melanophores has been deduced: in the absence of xanthophores, the dense iridophore regions of the light stripe expand and ingress into the stripe region, breaking it up into spots (*Frohnhofer et al., 2013*). Therefore, we assume that xanthophores mutant for *leo* and *luc* have lost their ability to restrict the expansion of dense iridophores. This explains the reduction of blue iridophore regions in the *leo^tK3;nac* double mutants, when compared with *nac* alone.

A requirement for *leo* and *luc* in melanophores, as observed in the chimeras (see below, *Figure 6C*), is corroborated by the phenotypes of the double mutants with *pfe*: if *leo* and *luc* were only required in xanthophores, the phenotypes of *pfe* mutants, which lack xanthophores completely, and the double mutants should be identical. However, we find an even distribution of melanophores in the double mutants rather than the series of spots that are still present in *pfe* (*Figure 5C,D*). This phenotype suggests a requirement of *leo* and *luc* for the homotypic association of melanophores.

## Cx41.8 and Cx39.4 channels are required in xanthophores and melanophores but not in iridophores

Chimeric animals created by blastomere transplantations suggested that the function of *leo* is required in melanophores and in xanthophores (*Maderspacher and Nusslein-Volhard, 2003*). The requirement in iridophores had not been tested. We assessed the requirement in all three chromatophore types by creating chimeric animals with the allele *leo^tK3*, where all *leo* and *luc* function is absent. In these experiments, we used appropriate combinations of mutant donor and host embryos to assert that in the chimeric animals only one of the three chromatophore types was mutant. First, we tested whether *luc* and *leo* have any function in iridophores. Therefore we transplanted cells from *nac;pfe;leo^tK3* donors into *rse* hosts (that lack iridophores). The donor embryos, due to mutations in *nac* and *pfe,* provide mutant iridophores as the only chromatophore type, whereas the melanophores and xanthophores of the chimeric animals must result from the *rse* mutant host that can form only rudimentary stripes. Thus, chimeric animals develop with *leo^tK3* mutant iridophores juxtaposed to wild-type xanthophores and

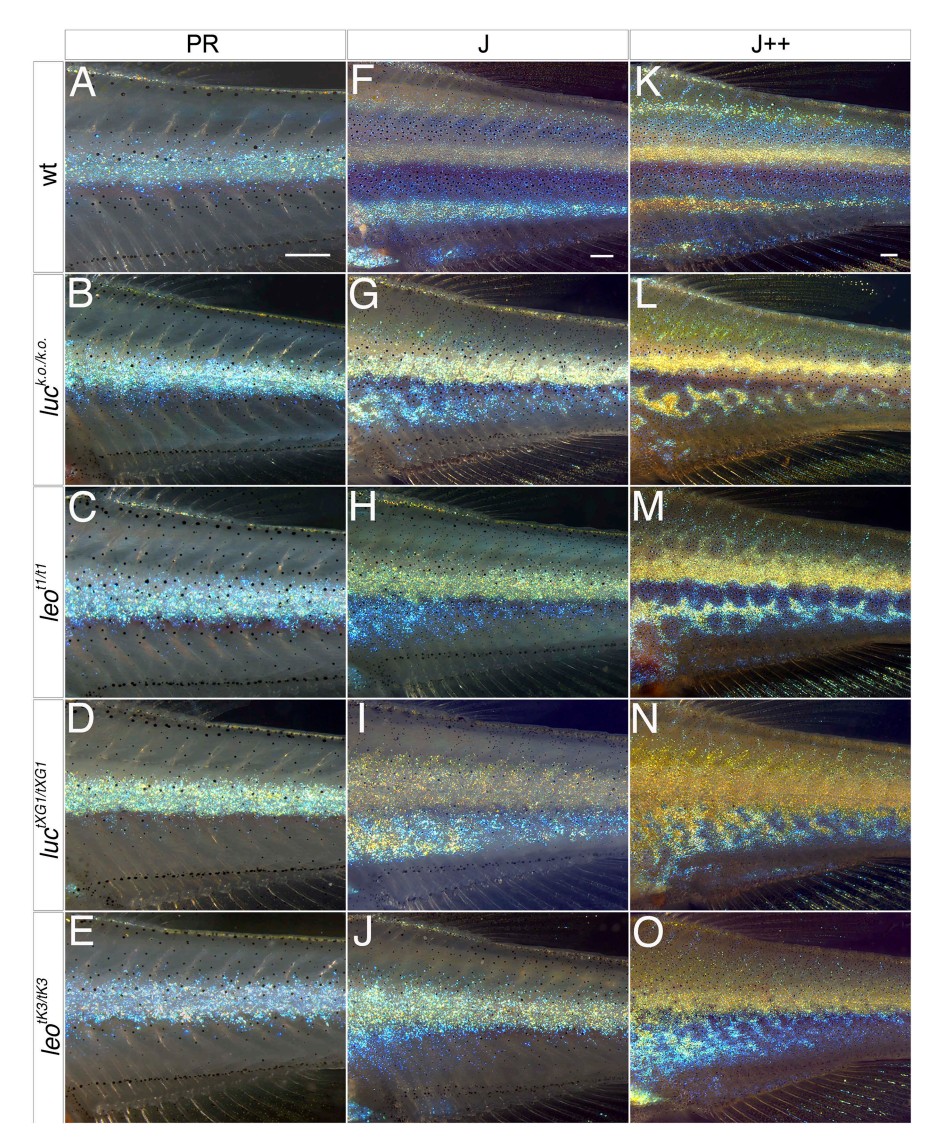

**Figure 4**. The *leo* and *luc* phenotypes arise during metamorphosis. In wild-type dense iridophores of consecutively formed interstripes become visible during stages PR (**A**), J (**F**), and J++ (**K**). Melanophores appear in the stripe regions. In *luc* loss-of-function (**B**, **G**, **L**), *leo*$^{t1}$ (**C**, **H**, **M**), *luc*$^{tXG1}$ (**D**, **I**, **N**), and *leo*$^{tK3}$ mutants (**E**, **J**, **O**) iridophores of the first interstripe form normally at stage PR (**B**–**E**), however, the melanophores ventral to the first interstripe are more scattered. In the mutants at stage J (**G**–**J**) the second interstripe fails to form in the ventral region where dense iridophores appear in irregular positions. Later at stage J++ irregular stripes and spots become visible in *luc* loss-of-function and *leo*$^{t1}$ mutants (**L** and **M**), in *luc*$^{tXG1}$ and *leo*$^{tK3}$ mutants (**N** and **O**) the number of melanophores is low and dense iridophores cover increasing areas on the flank of the fish. Scale bars: 0.25 mm.

melanophores. In these animals, clusters of donor derived mutant iridophores can cause a complete restoration of the stripe pattern (*Figure 6A,A'*), demonstrating that the function of *leo* and *luc* is not required in iridophores. Next, we tested the requirement for *leo* and *luc* function in xanthophores. Therefore we created chimeric fish with mutant xanthophores next to wild-type melanophores by transplanting cells from *leo*$^{tK3}$;*spa;rse* mutant blastula embryos (that provide mutant xanthophores [*Parichy et al., 1999*]) into *pfe* hosts (lacking xanthophores). Whereas the transplantation of wild-type xanthophore progenitors can completely restore the striped pattern (*Parichy and Turner, 2003*), the chimeric animals that received *leo*$^{tK3}$ xanthophores display a distribution of melanophores strongly resembling the *leo*$^{tK3}$ mutant phenotype, indicating that mutant xanthophores cannot restore the wild-type

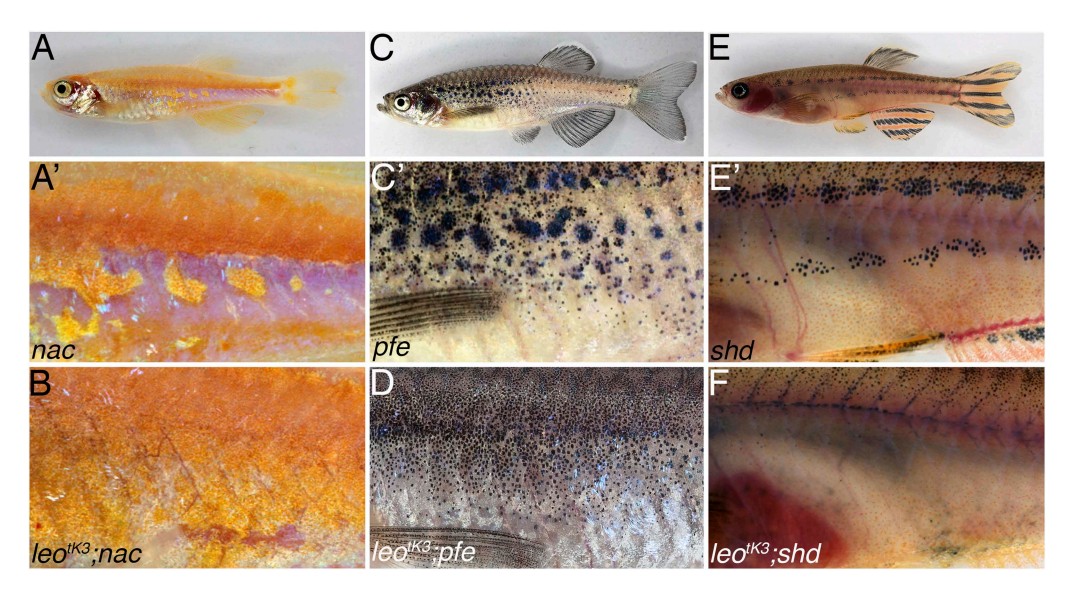

**Figure 5**. Complete loss of the pigment pattern in double mutants of *nac*, *pfe*, and *shd* with *leo^tK3^*. Mutants lacking one pigment cell type, *nac* (**A** and **A'**), *pfe* (**C** and **C'**), and *shd* (**E** and **E'**), only form remnants of the pigmentation pattern. In *nac* mutants fields of dense iridophores covered with compact xanthophores (corresponding to light stripes) and loose iridophores (corresponding to dark stripes) are discernible as patches (**A'**). In homozygous double mutants *leo^tK3^;nac* (**B**) the regions of dense iridophores covered by compact xanthophores expand at the expense of blue iridophore areas. In *pfe* mutants (**C**) in the absence of xanthophores the melanophores form patches in the regions around the first light stripe but are increasingly more disorganized at some distance to it (**C'**). In double mutants of *pfe* with homozygous *leo^tK3^* melanophores are distributed almost evenly as individual cells over the flank of the fish (**D**). In *shd* mutants (**E**), where no iridophores are present, melanophores and xanthophores only partly form the first light and dark stripes (**E'**). In double homozygous double mutants *leo^tK3^;shd* this residual pattern is lost and only very few individual melanophores are left on the flank of the fish (**F**).

The following figure supplement is available for figure 5:

**Figure supplement 1**. Double mutants of *nac*, *pfe*, and *tra* with *leo^t1^* and *luc^k.o.^*.

striped pattern (*Figure 6B,B'*). Therefore the *leo;luc* function must be required cell autonomously in xanthophores. Finally, we assessed the function of *leo* and *luc* in melanophores by transplanting cells from *leo^tK3^;pfe;rse* donor embryos into *nac* hosts. In these cases, where mutant melanophores are confronted with non-mutant xanthophores and iridophores, we again did not get a normal stripe pattern. However, the mutant melanophores did cluster together producing meandering fields of cells rather than creating spots or staying as individual cells, like in *leo^tK3^* (*Figure 6C,C'*). This indicates that the function of *leo* and *luc* is required in melanophores, but the wild-type xanthophores are able to exert some normalising influence over the mutant melanophores. Similar transplantation experiments using *leo^t1^* or *luc^tXG1^* embryos instead of *leo^tK3^* confirm that *leo* and *luc* are required in xanthophores and, to a lesser extent, in melanophores, but not in iridophores (*Figure 6—figure supplement 1*). Taken together, heteromeric connexons formed by Cx41.8 and Cx39.4 are not required in iridophores, but the function of *leo* and *luc* is required in melanophores and xanthophores instructing iridophores.

## Discussion

### A new connexin involved in pigment pattern formation

The *leo* phenotype varies, depending on the strength and combination of different alleles, from irregular, wavy stripes that are occasionally interrupted, over rows of spots to a complete loss of the pigment pattern, where isolated melanophores remain dispersed in a uniform background of dense iridophores and xanthophores. In the complete absence of Cx41.8 protein, in *leo^t1^* mutants, there is still some patterning activity present, leading to the formation of spots. The homozygous *leo^tK3^* phenotype is

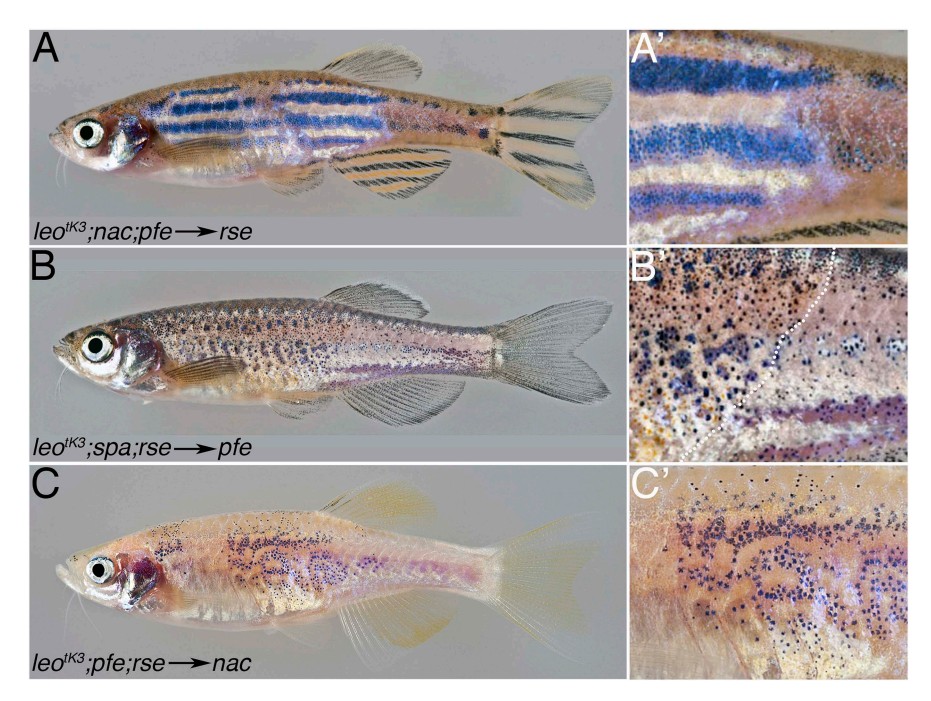

**Figure 6**. *leo* gap junctions are required in melanophores and xanthophores but not in iridophores. Typical examples of chimeric animals derived from blastomere transplantations are shown. Cells from *leo^tK3*;*nac*;*pfe* donor embryos, which can provide iridophores as their only chromatophore type, were transplanted in *rse* hosts, which lack most iridophores. The resulting chimeric animals (17 clones in 34 fish) show that mutant iridophores form a wild-type pattern when confronted with wild-type xanthophores and melanophores (**A** and **A'**). When cells are transplanted from *leo^tK3*;*spa*;*rse* donors into *pfe* hosts, thereby creating chimeras with mutant xanthophores (present in the anterior part of the fish, left to the dotted line in **B'**) and wild-type iridophores and melanophores, there is no wild-type pattern formed, but the chromatophores are almost uniformly distributed (**B** and **B'**) (9 clones in 72 fish). When cells are transplanted from *leo^tK3*;*pfe*;*rse* donors into *nac* hosts, thereby creating chimeras with mutant melanophores and wild-type iridophores and xanthophores (3 clones in 68 fish), again, no wild-type pattern is formed, but the mutant melanophores form loosely arranged fields of cells (**C** and **C'**).

The following figure supplement is available for figure 6:

**Figure supplement 1**. *leo* and *luc* are required in xanthophores and melanophores.

considerably stronger than this, indicating that the mutant Cx41.8 protein present in *leo^tK3* cells has a dominant negative effect on some further component(s) involved in pigment pattern formation. In a simple model, this additional factor could be another connexin involved in the formation of hetero-meric gap junctions together with Cx41.8, a notion that has been suggested previously, based on the analysis of *leo^tq270* (**Watanabe et al., 2006**). The zebrafish genome contains 37 genes coding for con-nexins and it is impossible to deduce from the predicted amino acid sequences alone possible partners for Cx41.8. Therefore we performed a mutagenesis of *leo^t1* fish and screened for mutants enhancing the phenotype to a strength similar to *leo^tK3*. In this screen we identified two dominant alleles of *luc*; we show that *luc* encodes another connexin, Cx39.4, and that loss-of-function of *luc* leads to a pheno-type very similar to that of *leo^t1*. The loss of both connexins results in a phenotype that is as strong as the one seen in homozygous mutants for the dominant alleles of *leo* or *luc*. Therefore the strong phenotypes of the dominant alleles can be fully explained by the loss of the two proteins.

Cx39.4 is a fish specific connexin, closely related sequences can only be found in the genomes of other teleost fish. It has been classified either as novel (**Eastman et al., 2006**) or placed into an orthol-ogy group together with mammalian Cx37 (GJA4) (**Cruciani and Mikalsen, 2006**) with which it shares 54% amino acid similarity. However, Cx39.4 is the only α-Connexin where the N-terminal cytoplasmic domain of 23 amino acids is extended by two residues. As in the case of Cx41.8, the N-terminal

cytoplasmic domain in Cx39.4 also contains an ExxE motif, therefore the conductivity of gap junctions formed by Cx39.4 could also be regulated by polyamines (*Musa et al., 2004*; *Watanabe et al., 2012*). Both, Cx41.8 and Cx39.4, have neither an R type nor a W type sequence motifs regulating heteromeric compatibility (*Koval et al., 2014*), so that no sequence based predictions about their ability to form heteromeric complexes are possible.

## Requirement for *leo* and *luc* function in chromatophores

Based on transplantation experiments, the function of *leo* was reported to be required in melanophores and in xanthophores (*Maderspacher and Nusslein-Volhard, 2003*). We repeated the experiments with the loss-of-function allele *leo^t1^*, the new strong allele *leo^tK3^*, and one of our dominant alleles of *luc* as donors. In agreement with the previous study, we show that *leo* function is needed in melanophores and xanthophores. We find the same requirement for *luc* function in both cell types. But neither *leo* nor *luc* has a role in iridophores. The results of our transplantation experiments also show that there is no contribution of non-pigment cells to the *leo* and *luc* phenotypes. Otherwise we would not have expected to find the complete rescue of a wild-type pattern to coincide in all cases with the transplanted iridophores into the *shd* or *tra* mutants.

Mutant melanophores, in chimeric animals, seem to be influenced by wild-type xanthophores in their neighbourhood, as they form continuous meandering patches larger than the spots found in *leo* or *luc* mutants. However, unlike wild-type melanophores when transplanted into *nac* hosts (*Maderspacher and Nusslein-Volhard, 2003*), they do not form normal stripes, so clearly *leo* and *luc* are required in melanophores for the formation of a wild-type pattern. The function of both genes is also required in xanthophores; mutant xanthophores, when confronted with wild-type melanophores in the chimeras, do not generate a wild-type pattern but spot or disperse melanophores in large dense iridophore regions. When we tested iridophores we found that they require neither *leo* nor *luc* function, as the mutant cells can contribute to a completely wild-type pattern if they are next to other wild-type cells. This is surprising and indicates that the dramatic expansion of the dense iridophores that is associated with the mutant phenotypes is an indirect effect, possibly caused by mutant xanthophores covering the iridophores.

## Gap junction communication between chromatophores

Taken together our data suggest that Cx41.8 and Cx39.4 interact to form heteromeric gap junctions and that these channels are responsible for some of the interactions that take place among chromatophores while the pigment pattern is established. This is corroborated by our analysis of double mutants where one chromatophore type is missing and *leo* or *luc* is mutant. In the absence of xanthophores, in *pfe* mutants, melanophores cluster together; this clustering is lost in double mutants with *leo^tK3^*, where *leo* and *luc* function are affected, indicating that the channels are responsible for homotypic interactions among melanophores. In the double mutants of *leo^tK3^* with *nac*, where melanophores are missing, mutant xanthophores are unable to confine the light stripe regions, suggesting an instructive interaction between xanthophores and iridophores during normal development. In double mutants with *shd* or *tra*, where iridophores are lacking, the number of melanophores is further reduced and xanthophores are evenly distributed, indicating that in wild-type there might be a positive interaction between melanophores and xanthophores via gap junctions. The phenotypes of the double mutants with the *leo* or *luc* loss-of-function alleles are generally a bit weaker than with *leo^tK3^*. This indicates that in the absence of one of the connexins, when heteromeric channels cannot form, homomeric channels retain some functionality.

The *leo* and *luc* mutant phenotypes seem to be caused by iridophores not behaving appropriately and failing to undergo the transition from the dense to the loose form at the correct time and place. Recently it was shown that proliferation and dispersal of iridophores during metamorphosis, the time when the adult pigment pattern is generated, outline the light and dark stripe regions in the trunk of zebrafish (*Singh et al., 2014*); at the same time the larval xanthophores also start to proliferate, they re-organize and change their shapes to densely cover the light stripe regions and loosely the dark melanophore stripes (*Mahalwar et al., 2014*). This raises the possibility that in normal development the xanthophores, which cover the iridophores, induce the transition to the loose form when they reach the regions destined to form dark stripes. It is interesting in this context that in *schachbrett* (*sbr*) mutants, which display interrupted dark stripes, the affected gene is specifically expressed in iridophores and codes for Tight Junction Protein 1a (Tjp1a, ZO-1) (Andrey Fadeev, JK, HGF, and CN-V, submitted). If gap junctions are indeed involved in the signalling events between xanthophores and iridophores,

there must be (at least) one, so far unidentified, connexin, which is expressed in iridophores. Alternatively, other signalling pathways could be responsible for the cross-talk between iridophores and xanthophores, and gap junctions might only be required for the homotypic interactions among xanthophores. Whether direct interactions between xanthophores and melanophores, which have been demonstrated in an in vitro system (*Yamanaka and Kondo, 2014*), are also possible in vivo is not known. In the skin of the fish these two cell types are always separated by a layer of iridophores (*Hirata et al., 2003*, *2005*; *Watanabe et al., 2006*), however, recently it has been shown that melanophores of the dark stripe produce long cellular extensions reaching towards the light stripe xanthophores (*Hamada et al., 2014*); these extensions could allow direct contact between melanophores and xanthophores. These direct cell-to-cell contacts would permit communication between the cells via gap junctions.

Our data suggest that in the absence of one connexin-partner, homomeric channels can exist, but they retain only some residual functionality, as can be seen in *leo* or *luc* loss-of-function mutants. The non-functional proteins present in the dominant mutants can lead to a complete block of activity, equal to the one seen in the *leo*[k.o.];*luc*[k.o.] double mutants. Interestingly, in *luc* mutants the fins are striped, suggesting that homomeric *leo* channels suffice for patterning the fins, where stripes can be formed in the absence of iridophores.

## Materials and methods

### Fish husbandry

Zebrafish were maintained as described earlier (*Brand et al., 2002*); we used the following genotypes: Tübingen, *leo*[t1], *nac*[W2], *pfe*[tm236b], *rse*[tLF802], *shd*[j9s1], *spa*[b134], *tra*[b6]. ENU mutagenesis was performed as described (*Rohner et al., 2011*), 40 male fish were mutagenized over the course of 6 weeks. All animal experiments were performed in accordance with the rules of the State of Baden-Württemberg, Germany, and approved by the Regierungspräsidium Tübingen (Aktenzeichen: 35/9185.81-5/Tierversuch-Nr. E 1/09).

### Genotyping

For the PCR amplification of genomic DNA from fin clips, the tissue samples were incubated in TE supplemented with 5% Chelex-100 (BioRad Laboratories, Hercules, CA) and 10 µg/ml Proteinase K (Roche, Germany) for 4 hr at 55°C and 10 min at 95°C and then stored at 4°C. 0.1–1 µl of the supernatant was used as template in a standard PCR of 25 µl. The following oligonucleotides were used as primers:

leo_for1: 5′-CTCTTATTTCCACCTCAGGCTCC-3′
leo_for2: 5′-GCGGTGTCACTGCTGCTTAGTATC-3′
leo_rev1: 5′-CTATTCGACTGCATGAAGGTTGC-3′
leo_rev2: 5′-CAACTGCTTCCATGCCAAATG-3′
obe_for1: 5′-GAAACTATTCTTGCCGTGACTTGG-3′
obe_for2: 5′-TGTTCACAGTGACATGCACAGATG-3′
obe_rev1: 5′-TCAAACAAACCTGGGTGTGGAC-3′
obe_rev2: 5′-GTGTACCTAACAGGGCAAACGG-3′
luc_for1: 5′-TGCCTCTAGGAACATGATTGGG-3′
luc_rev1: 5′-TCAAACATAATGTCTCGGTTTG-3′
luc_rev2: 5′-CGGAGTTGAGACGAAGATGACC-3′

### Generation of *luc* k.o.

The *luc* loss-of-function was generated using the CRISPR-Cas system as described in *Hwang et al. (2013)*. The following oligonucleotides were cloned into pDR274 to generate the sgRNA vector:

Cx39.4_1: 5′-TAGGAAGGCCAGGAGTACTCGA-3′
Cx39.4_2: 5′-AAACTCGAGTACTCCTGGCCTT-3′

One-cell stage embryos were injected with 2–4 nl of a solution containing 200 ng/µl Cas9 mRNA and 15 ng/µl sgRNA. Cas9 mRNA was transcribed from the linearized plasmid pMLM3613 using the mMessage mMachine Kit and a polyA tail was added with the Poly(A) Tailing Kit (both Ambion(Europe), UK). The sgRNA was transcribed from the linearized vector using the Megascript Kit (Ambion). The efficiency of indel generation was tested on eight larvae at 4 dpf by PCR and sequence analysis. The other injected larvae were raised to adulthood and mature $F_0$ fish with pigmentation pattern defects were incrossed. $F_1$ fish from these crosses were genotyped and fish with 2 bp and 22 bp deletions were used to establish mutant stocks.

## Blastula transplantations

Chimeric animals were generated by transplantations of cells during blastula stage as described (*Kane and Kishimoto, 2002*).

## Image acquisition and processing

Adult fish were anaesthetised with 0.004% MS-222 (Sigma-Aldrich, Germany) and photographed with a Canon EOS 5D MarkII camera and a Macro 100 objective. Metamorphic fish (*Figure 4*) were anaesthetised, embedded in 1.5% low melting point agarose with 4.5 mg/ml ± epinephrine (Sigma-Aldrich) for melanosome contraction, and photographed under a Leica M205 FA stereomicroscope with a Leica DCF300 FX camera using the software LAS V4.1 to allow multifocus images.

## Acknowledgements

We thank Brigitte Walderich for her support with the zebrafish work, Christian Söllner, Prateek Mahalwar, and Ajeet Singh for many insightful discussions, and Ajeet Singh for his comments on the manuscript.

# Additional information

## Funding

| Funder | Grant reference number | Author |
| --- | --- | --- |
| European Commission | ZF-HEALTH grant, 242048 | Christiane Nüsslein-Volhard |

The funders had no role in study design, data collection and interpretation, or the decision to submit the work for publication.

## Author contributions

UI, HGF, Conception and design, Acquisition of data, Analysis and interpretation of data, Drafting or revising the article; JK, Conception and design, Acquisition of data, Analysis and interpretation of data; TÇC, H-MM, SG-R, CW, Acquisition of data, Analysis and interpretation of data; CN-V, Conception and design, Analysis and interpretation of data, Drafting or revising the article

## Ethics

Animal experimentation: All animal experiments were performed in accordance with the rules of the State of Baden-Württemberg, Germany. The protocol for ENU mutagenesis was approved by the Regierungspräsidium Tübingen (Aktenzeichen: 35/9185.81-5/Tierversuch-Nr. E 1/09).

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
