## [Decision Letter]

Thank you for sending your work entitled “Gap junctions composed of connexions 41.8 and 39.4 are essential for colour pattern formation in zebrafish” for consideration at *eLife*. Your article has been favorably evaluated by Diethard Tautz (Senior editor), a Reviewing editor, and 2 reviewers.

The following individuals responsible for the peer review of your submission have agreed to reveal their identity: Marianne Bronner, BRE; Rob Cornell and Richard White, peer reviewers

The Reviewing editor and the two reviewers discussed their comments before we reached this decision, and the Reviewing editor has assembled the following comments to help you prepare a revised submission.

This paper contributes to a growing literature that uses stripe formation as a model for cell interactions during organogenesis. Irion and colleagues use forward and reverse genetic methods in zebrafish to identify a new gene that contributes to stripe formation. By performing a genetic modifier screen in the *leo* (t1) background, they identified two alleles of cx39.4, which they refer to as luchs. Using transplantation and mutant crosses, they propose a model where cx39.4 and cx41.8 form a heteromeric gap junction required in xanthophores and melanophores, although there are no biochemical experiments to prove this. They conclude that xanthophores mutant for the two channels lack the ability to restrict the expansion of dense iridophores (iridophores are otherwise confined to light stripes) resulting in the visual phenotype of light stripes expanded into the dark stripes. In addition the channels are necessary within melanophores for the homotypic clustering of melanophores in the absence of xanthophores.

A particularly elegant aspect of this work is the use of a dominant enhancer screen in zebrafish to find the protein that interacts connexins 41.8 (encoded by the *leo* locus) whose existence had been implied by the observation that the dominant alleles of *leo* had a stronger phenotype than the null alleles. There are very few other published examples of enhancer and suppressor screens using zebrafish. Another strength is the construction of appropriate triple mutants that establish that melanophores and xanthophores, but not iridophores, require connnexins 41.8 and 39.4 autonomously for the proper organization of the iridophores.

The following additional experiments would strengthen the paper:

1) The authors conclude that there is “no contribution of non-pigment cells to the *leo* and *luc* phenotypes”, which is supported by their transplant data. However, it would be very useful to characterize the expression pattern of cx39.4 during embryonic development, either through in situ or transgenic reporters. This would support the model that the gene is required in these cell types, but would also give insight into additional cell types that may not be phenotypically obvious but still dependent upon this protein.

2) Is it known whether *luc* deficient precursors have a survival vs. differentiation defect? Were the chimeric transplants (i.e. *nac;pfe;leo-tk3*) done with GFP labelled blastula cells, which could help discern whether those cells were even present when the adults were assayed?

3) There is no mention of numbers of animals or any statistical considerations at all. At a minimum, this needs to be provided for the blastula transplant experiments, which are often somewhat noisy, yet many of the conclusions of the paper are based on this. In general, a more clear quantification of some data would make some conclusions more convincing.

4) The Introduction is needlessly long and could be shortened by 1/3 to 1/2 of its current length.

---

## [Author Response]

*1) The authors conclude that there is “no contribution of non-pigment cells to the leo and luc phenotypes”, which is supported by their transplant data. However, it would be very useful to characterize the expression pattern of cx39.4 during embryonic development, either through in situ or transgenic reporters. This would support the model that the gene is required in these cell types, but would also give insight into additional cell types that may not be phenotypically obvious but still dependent upon this protein*.

We agree that it would be very good to know where the gene is expressed, not only during embryonic development, but also during metamorphosis, the stage when the adult pigmentation pattern is established. Through RT-PCR experiments we know that *cx39.4* mRNA is present in embryos and in light and dark stripes of adult fish. Unfortunately, however, we were so far unable to obtain clear and unambiguous in situ signals; this might possibly reflect low levels of expression in the embryo and the technical difficulties of in situ hybridizations during later stages.

Also, we agree that a reporter transgene would be very useful, however the construction of one is hampered by the fact that the promoter of *cx39.4* is entirely uncharacterized. The gene has an intron in the 5’ untranslated region of 7.7 kb and an upstream (gene-less) region of approx. 500 kb, making BAC recombineering or, preferably, knock-in by homologous recombination the methods of choice for the construction of such a reporter. Both methods are rather time-consuming and labour intensive and it is not possible to get results within the next few months.

With the experiments described in the manuscript we have shown that the function of both connexins is required in xanthophores and in melanophores. Although we have no direct proof, we think it is unlikely that other cell types are dependent upon the proteins for two main reasons. First, we have hypermorphic alleles for both genes; these could uncover functions of the proteins that might otherwise be masked by redundancy, but mutant fish are viable and fertile, and no additional phenotype could be detected. Second, in the transplantation experiments of *leo*^*tK3*^*;nac;pfe* into *rose*, where we used donors that provide iridophores as the only type of chromatophore, we obtained good rescue in all chimeras. It is very likely that in several of these cases we also transplanted other cells, however they never lead to phenotypic consequences.

*2) Is it known whether* luc *deficient precursors have a survival vs. differentiation defect? Were the chimeric transplants (i.e.* nac;pfe;leo-tk3*) done with GFP labelled blastula cells, which could help discern whether those cells were even present when the adults were assayed?*

There are probably two different issues here; first, the number of melanophores is reduced in mutants homozygous for the dominant *leo* or *luc* alleles. In these cases it is interesting to ask whether this is due to survival or differentiation defects; however it is not easy to obtain unambiguous answers, using for example *mitfA:GFP* as a marker, and we haven’t done the experiment, so far.

What we can say is, that the defect is not autonomous to melanophores, but depends on the presence of xanthophores and therefore presumably on cell-cell interactions, because the melanophore number is greater in *leo*^*tK3*^*;pfe* double mutants compared to *leo*^*tK3*^ alone.

The second issue is about the transplantations we present in our manuscript; in the case of *leo*^*tK3*^ we used triple mutant donors, which can only provide one type of chromatophore, and transplanted cells into hosts lacking exactly this cell type. Therefore we can be sure that the cells we identify later in the chimeric animals (due to their pigment content) are indeed derived from the donor and therefore carry the *leo* mutation. Nevertheless, transplanting GFP-labelled cells would also have been informative, e.g. in providing information about transplanted non-pigment cells. However it takes at least two generations, i.e. a minimum of six months, to get a GFP-transgene into the triple mutant background. Without one further generation of inbreeding only 3 in 256 donor fish will have at least one copy of the transgene and be homozygous for the three mutations; therefore we did not carry out these experiments.

*3) There is no mention of numbers of animals or any statistical considerations at all. At a minimum, this needs to be provided for the blastula transplant experiments, which are often somewhat noisy, yet many of the conclusions of the paper are based on this. In general, a more clear quantification of some data would make some conclusions more convincing*.

We have added the number of fish and the number of clones obtained for each of the transplantation experiments in the figure legends. Any additional statistical evaluation seems inappropriate, given the small numbers and the variable sizes and positions of the clones

*4) The Introduction is needlessly long and could be shortened by 1/3 to 1/2 of its current length*.

We have shortened the Introduction to 1062 words.